# The presence of Wormian bones increases the fracture resistance of equine cranial bone

**Lilibeth A. Zambrano M.**[1], **David Kilroy**[2], **Arun Kumar**[2], **Michael D. Gilchrist**[1], **Aisling Ní Annaidh**[1] *

1 School of Mechanical and Materials Engineering, University College Dublin, Dublin, Ireland, 2 School of Veterinary Medicine, University College Dublin, Dublin, Ireland

* aisling.niannaidh@ucd.ie

**Data Availability Statement:** All relevant data are within the manuscript and its Supporting information files.

## Abstract

Wormian (intrasutural) bones are small, irregular bones, that are found in the cranial sutures of the skull. The occurrence of Wormian bones in human skulls has been well documented but few studies have detected the presence of such bones in domestic animals. Although some research has linked the presence of Wormian bones to bone pathology, its anatomical significance in healthy individuals is not known. To the best of our knowledge, no previous study has examined the biomechanical features of Wormian bone. This study uses microCT imaging of the parietal bone region to determine the frequency of occurrence of Wormian bones in horse skulls and, through 3-point bending tests, to calculate the mechanical differences that result from the presence of such bones. In addition, bone properties such as bone mineral density (BMD) and stiffness were measured and analysed to determine the influence of Wormian bone. Our findings on 54 specimens taken from 10 horses (ages ranging from 4 to 29 years) showed that Wormian bone was present in 70% of subjects and that its occurrence was unrelated to age or sex. 3-point bend tests revealed that the stiffness normalised by cross section area (P = 0.038) was lower in samples where Wormian bone was present. An idealised Finite Element simulation confirmed that the presence of Wormian bone reduced the maximum stress and strain, as well as their distribution throughout the sample. We consequently conclude that the presence of Wormian bones, which are confined to the calvaria, increase the compliance of the bone and reduce the likelihood of skull fracture. As all skull samples were collected from a local abattoir, ethical approval was not required for this work.

## Introduction

Wormian bones are irregular bones found within the sutures of the skull (Fig 1). They are usually located in the cranial sutures of the dorsal and caudal regions of the calvaria (braincase). The first reference to their occurrence in the human skull is attributed to the Swiss physician of the German Renaissance, Paracelsus, and these bones are named after the 17th-century Danish anatomist, Olaus Wormius [1]. Anthropological studies on human skulls have found Wormian bones in most of the specimens reviewed, varying from 55% in an Anglo-Saxon

**Funding:** LZ was funded by the European Union's Horizon 2020 research programme under the Marie Sklodowska – Curie grant agreement No. 642662. The funders had no role in study design, data collection and analysis, decision to publish, or preparation of the manuscript.

**Competing interests:** The authors have declared that no competing interests exist.

**Fig 1. Illustration of Wormian (intrasutural) bones located in the lamboid suture in the human skull (indicated by asterisks and circled); caudal view of the skull.** Lp: Left parietal bone; Rp: Right parietal bone; Oc: Occipital bone (modified from [6]). Used with permission.

population to 80% in a Chinese population [2]. Likewise, animal studies have recorded a varying incidence of these bones, varying from 22% (rodent species) to 58% of hedgehogs [3]. It is clear that the incidence of Wormian bones varies across populations and this may be an indication that its prevalence exhibits genetic variation [4, 5].

While numerous studies have investigated the incidence of Wormian bones and their location within the skull sutures [1, 2, 7], relatively little research has discussed the functional significance of these bones. Most of the early research [1] comment on the function of Wormian bones, claiming that these bones provide supplementary areas of ossification which cover and protect the cranial cavity. Some researchers have suggested a pathological significance for the presence of Wormian bones, especially the condition osteogenesis imperfecta [8, 9]. Others have suggested a link between infant skull compression (due to cradling the skull, leading to greater pressure on the caudal portion of the calvaria) and an increased incidence of Wormian bones [10, 11]. In contrast, a comparative study of the incidence of Wormian bone in skulls with deformities and in normal skulls [12, 13] did not find any statistical difference between the two groups.

Apart from Pucek [3], little work has been conducted on the presence of Wormian bones in domestic animals. Consequently, we considered the horse as a suitable species to study as it has a well-developed interparietal bone and the condition of osteogenesis imperfecta has not been reported in this species [14]. As far as the authors have been able to ascertain, no previous studies have investigated the influence of Wormian bones and only a few studies have investigated the influence of sutures on the distribution of strain and stress within an animal's skull [15–17] as a consequence of mechanical load.

Both finite element (FE) and model-based design (MBD) techniques have previously been employed to simulate the full skull structure under different loading conditions associated with diverse biting regimes [18], proving that sutures are associated with stress and strain transfer in the lizard and pig. However, unlike load bearing bones, such as the limbs and jaw, the cranium in mammals does not experience significant in vivo bone strain and does not undergo load-induced bone formation [19]. Previous authors have hypothesised that not all bones develop through 'functional adaption' and this leads to the strength of the cranium comfortably exceeding the upper limit of functional strains associated with biting or chewing [20]. Instead, the cranium is hypothesised to be designed to counteract infrequent, traumatic loads to protect the underlying brain, perhaps through a natural selection process [21].

The present study uses microCT imaging of the parietal bone region to determine the occurrence of Wormian bones in horse skulls and, by means of 3-point bending tests and finite element simulations, quantifies the difference in bone strength that result from its presence. Our hypothesis is that the presence of Wormian bones increases the compliance and reduces the likelihood of skull fracture, potentially acting as an additional design feature that may exist to dissipate traumatic cranial loads and protect the underlying brain.

## Materials and methods

### Sample harvesting

The horse skull is comprised of 34 flat bones joined by sutures and can be divided into the cranium (*cranium cerebrale*) and the face (*cranium viscerale*). One of the bones in the *cranium cerebrale* is the parietal bone, which forms a dorsolateral wall protecting the brain [22] (Fig 2a). This bone presents a three-layered structured cross-section, composed of two external layers of cortical bone separated by an inner ply of trabecular or spongy bone (sandwich structure, Fig 2b).

Bone samples from equine parietal bones were collected from ten fresh-frozen heads (3 male and 7 female, 4–29 years old). These horses were obtained from an abattoir and were free from any metabolic or bone-related disease. Due to the origin of the samples, ethical approval was not required for this work.

After collection, the heads were stored at -20˚C for up to 2 weeks each. Specimens were defrosted for 12 hours until they reached 4˚C [23–26]. Using a scalpel, soft tissue was removed carefully from the bone. A vertical band saw was used to cut the parietal bone samples into

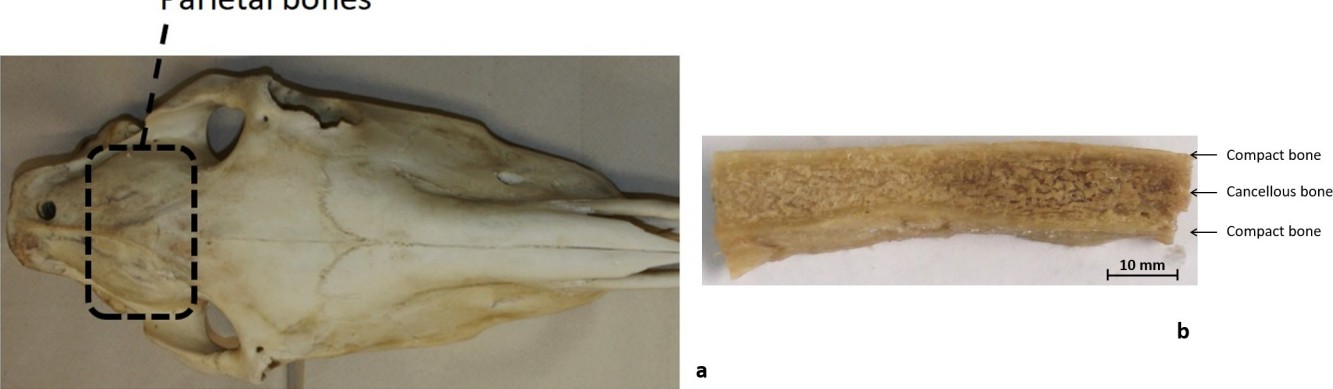

**Fig 2. Sample extraction from equine heads.** a) Dorsal view of horse head showing the location of the parietal bones b) Parietal bones beam-like sample.

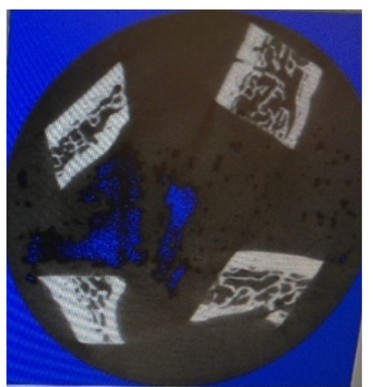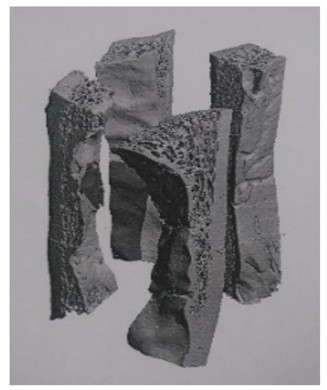

**Fig 3. Sample arrangement for microCT scans.** a) Cross-section image of the samples during the scan (midsection). b) 3D rendering of the scanned samples.

cuboid shapes, having average dimensions of 41.51 mm (±6.8 mm) x 9.48 mm (±1.61 mm) x 5.013 mm (±1.27mm) length x width x thickness (dx.doi.org/10.17504/protocols.io.xvkfn4w). A total of 59 specimens were obtained. The beam-like shaped samples were wrapped in gauze soaked in saline solution and refrigerated at 4˚C for a maximum of 12 hours [23–27], until screening was performed with a microCT scanner.

## Specimen scanning

A Scanco μCT40 Desktop cone-beam microCT scanner was used to capture microCT images of the parietal bone samples (70 kVp, 114 μA, 8 W), at a resolution of 36 μm. An average of four samples were placed inside a plastic tube (Fig 3a). All samples were immersed in saline solution to maintain hydration so that the bone properties were well preserved during the scanning process. The three-layered internal bone distribution of the parietal bone can be seen in Fig 3b. Average Bone Mineral Density (BMD) was determined for each sample directly from the microCT scans using Microview (Parallax Innovation).

CT scans of bone samples were examined to determine whether Wormian bones were present in the specimen (Fig 4a and 4b) and specimens were divided into two groups: those containing Wormian bones and those in which they were absent.

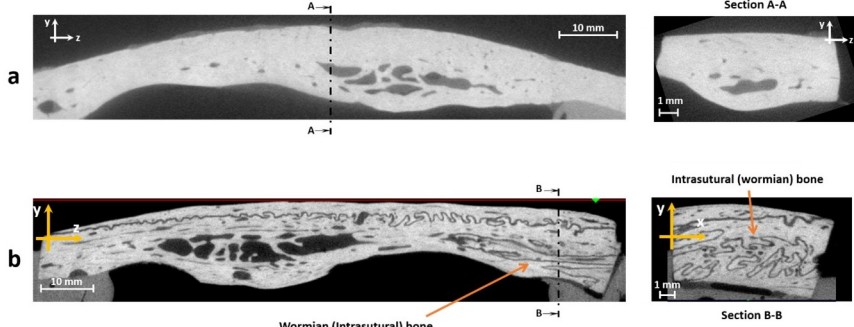

**Fig 4. microCT scans of equine parietal bone.** a) Lateral view (left) and cross-section (right) of a specimen without Wormian bone, b) Lateral view (left) and cross-section (right) of a sample with Wormian bone.

## Mechanical testing

Using a displacement-controlled loading procedure (Instron 8520 Servo-hydraulic dynamic testing system, load cell 5kN), the specimens were subjected to a 3-point bending test until failure (bone fracture) under quasi-static conditions (crosshead speed of 2.5 mm/s). As irregularities in the internal surface of the skull (bottom cortical layer) would cause misalignment of the samples during the load application, the ends of the specimens were embedded in epoxy, so that the top (outer) cortical layer of the specimens remained perpendicular to the probe during the loading process (Fig 5). The effective span length was thus reduced to 40 mm. All samples were kept hydrated with saline solution during this process.

The samples were tested according to the ASTM D7250/D7250M-16 standard [28] and force-displacement curves were recorded for each sample. From this data, the Young's modulus, $E$, and the structural (bending) stiffness, $S$ were calculated. The simplest way to calculate the bending stress and obtain mechanical properties of a straight beam is by using the Euler-Bernoulli formula, which establishes that the normal stress (tension or compression) is the main contributor to failure of the beam [29, 30]. In this case, the sample is a composite beam (comprising two different types of bone), therefore the Transformed Section Method was employed, whereby the composite beam can be treated as a single homogeneous beam [29, 30]. The Euler-Bernoulli theory can be applied (Eq 1) to calculate the bending stress, where $M$ is the bending moment acting on the sample, $c$ is the distance from the midline to the outer surface, and $I$ is the second moment of area.

$$\sigma = \frac{M*c}{I} \tag{1}$$

From here, the application of Hooke's law allows for the calculation of the elastic modulus E:

$$E = \frac{L^3 S}{4bt^3} \tag{2}$$

where $S$ (N/mm) represents the bending stiffness of the structure and is the slope of the force versus displacement curve [29, 30], $L$ represents the span length, $b$ the cross-sectional width and $t$ the cross-sectional thickness. The normalised bending stiffness NS (N/mm^3) is determined by dividing the bending stiffness by the average cross-sectional area of the sample.

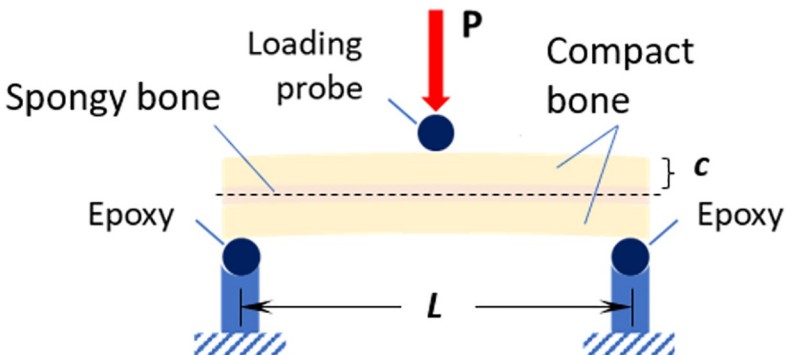

**Fig 5. Setup for three-point bending tests of equine parietal bone beam-like specimens, where $P$ is the applied load, $c$ is distance from the midline to the outer surface and $L$ is the span length.**

The calculation of the Young's modulus (Eq 2) is based on the assumption of a prismatic beam, a straight beam with constant cross-sectional area, which is not the case for the parietal bone specimens that have a very irregular and variable cross-section, as well as a curved shape. Before proceeding with the straight beam calculations, the straight beam criteria was employed. Under this criteria, the bending radius of the beam-like sample, $R$, is expected to be at least five times the thickness of the sample, $t$ [29]. All the samples in the current study complied with the straight beam condition and were used for the subsequent data analysis.

## Statistical analysis

Due to the low number of subjects tested (N = 10), a Fisher's exact test was used to evaluate the association between sex and the presence of Wormian bone, while a binary logistic regression was used to study the incidence of Wormian bone according to the age of the subject.

For the continuous variables of BMD, Stiffness $S$, *Normalised Stiffness NS* and Elastic modulus $E$, a normality test was performed before conducting a 2-sample t-test of the two groups of samples: with and without Wormian bone. The interdependence of samples (N = 56) was evaluated by calculating the interclass correlation coefficient. As the intra and inter-subject correlation was low (0.035 and 0.264 respectively, with p-values of 0.264 for both cases), samples were divided only according to the presence or absence of Wormian bone and the origin, or 'subject' from which the sample was extracted, was ignored. In this statistical test we were assessing the effect of the presence of Wormian bone on the mechanical behaviour of the sample, therefore the 'subject' is not relevant.

## Numerical simulations

Two idealised finite element models were created in Abaqus 2017 (Dassault Systémes Simulia Corp., Providence, RI, USA): one representing a sample with Wormian bone present, one representing a sample without Wormian bone. The purpose of these simulations was to confirm the effect of Wormian bone, while eliminating confounding factors which are inherently present when testing biological tissue (small sample size, heterogeneity, storage conditions etc.). As idealised models, they are not intended to capture the microscopic geometric features or properties of the samples, but to identify broad trends in macroscopic behaviour.

Both models consisted of a three-layered sandwich structure (Fig 6a), the first, representing a Non-Wormian bone sample (Fig 6b), was comprised of only cortical and trabecular bone. The second, representing a Wormian bone sample of the same thickness (Fig 6c), included a section of Wormian bone, represented as an additional undulating layer within the upper table of cortical bone. The models were evaluated in 3-point bending conditions, using simple supports at both ends and applying a displacement (1.4 mm) in the mid-span of the beam via a discrete rod (Fig 6a). A general contact condition between all the surfaces was chosen. Quadratic hexahedral elements were used throughout (C3D20R: a 20-node quadratic brick, reduced integration, hour-glass control, average size 1mm). Non-linear conditions for the displacements were considered in the analysis.

The geometric and material properties assigned to the models are shown in Table 1. Material properties for the compact and cancellous bone were applied from [32]. In the absence of material properties of Wormian bone available in the literature, the authors examined the density of this layer from the microCT scans. It is well established that the density of bone can be related to its Elasticity through well-known elasticity-density relationships [31]. S1 and S2 Figs compare the greyscale values of cancellous bone and Wormian bone samples. As the greyscale values (and therefore the density and elasticity) were similar, the same material properties were applied to both cancellous and Wormian bone.

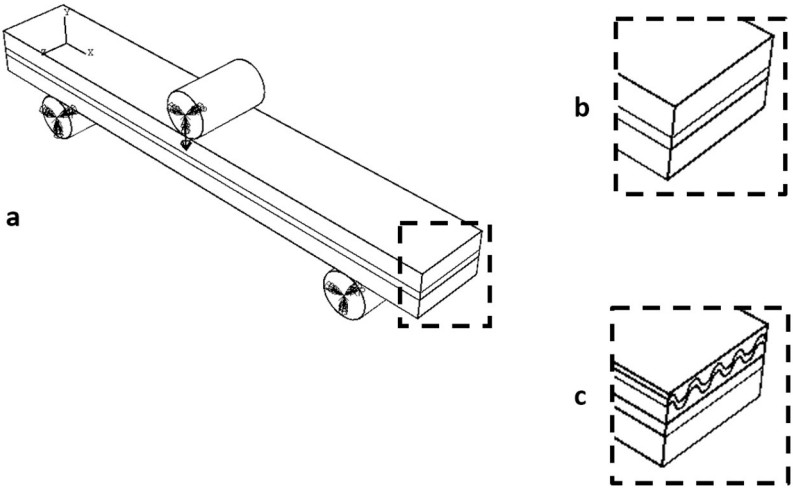

**Fig 6.** a) Idealised FE model representing the three-layered structure of the skull under 3-point bending, b) without Wormian bone, where a layer of spongy cancellous bone is sandwiched between to cortical layers, and c) including Wormian bone, where a Wormian layer is included as an additional undulating layer within the upper table of cortical bone.

**Table 1. Mechanical properties for idealised FE skull bone models [32].**

| Material | Elastic modulus (GPa) | Poisson ratio | Density (g/cc) | Layer Thickness (mm) |
|---|---|---|---|---|
| Compact bone | 20 | 0.21 | 1.800 | 2.5 |
| Cancellous bone | 4 | 0.21 | 1.000 | 1 |
| Wormian bone | 4 | 0.21 | 1.000 | 0.5 |

## Results

### Experimental results

As seen in Fig 7, no male subjects with Wormian bone were identified. However, from a Fisher's exact test, a p-value of 0.475 indicated that there is no apparent association between sex and Wormian bone presence in horse parietal bone. From a binary logistic regression, a p-

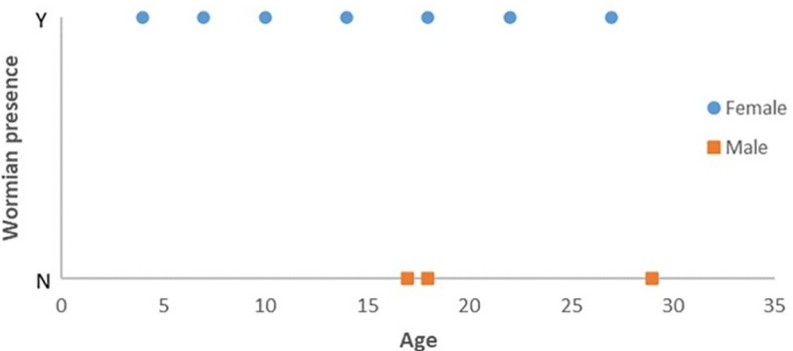

**Fig 7. Distribution of subjects according to sex, age and Wormian bone presence.** No trend is observed for any of these variables.

**Table 2. Mean values (SD) for bone mineral density BMD, deflection (w), strain rate ($\dot{\varepsilon}$), stiffness (S) and Young's modulus (E) for the two group of equine parietal samples (Wormian and non-Wormian) tested in 3point bending.**

|  | BMD (g/cm$^3$) | S (N/mm) | S normalised (S/mm$^2$) | E (GPa) |
|---|---|---|---|---|
| Wormian | 0.975 (0.147) | 147 (108) | 2.81 (1.29) | 2.23 (1.18) |
| Non-Wormian | 1.024 (0.174) | 179.9 (86.2) | 3.7 (1.84) | 2.67 (1.27) |
| p-value | 0.252 | 0.221 | 0.038 | 0.198 |

value of 0.872 indicated that the likelihood of Wormian bone being present is approximately equal at any age.

Three samples were discarded after testing due to slippage during the experiments so that results are based on a total of 56 successfully tested specimens. The mean values for BMD, bending stiffness (*S*), normalised bending stiffness *(NS)*, and Young's modulus (*E*) are presented in Table 2.

According to the t-test analysis, the values of BMD were not significantly affected by the presence of Wormian bone (p-value = 0.252, Fig 8a and Table 2). It should be noted, however, that the BMD is averaged over the entire volume of the sample, while the Wormian section may represent only a small percentage of the overall volume.

The bending stiffness of the Wormian group was ~22% smaller than the Non-Wormian group but the 2-sample t-test showed no statistical difference in bending stiffness values (p-value = 0.221). Nevertheless, this factor is also influenced by the geometry of the samples (stiffness is a structural property) whose cross-sectional area is highly variable. Hence, the bending stiffness was also normalised with respect to the average cross-sectional area of the specimen (see Fig 8c), which shows that the samples with Wormian bone (2.81 ± 1.29) are around 2% more flexible than those without Wormian bone (3.7 ± 1.84), [16–18, 33–35]. In this case, the t-test indicated a statistical difference (p-value = 0.038). The average Young´s modulus of the

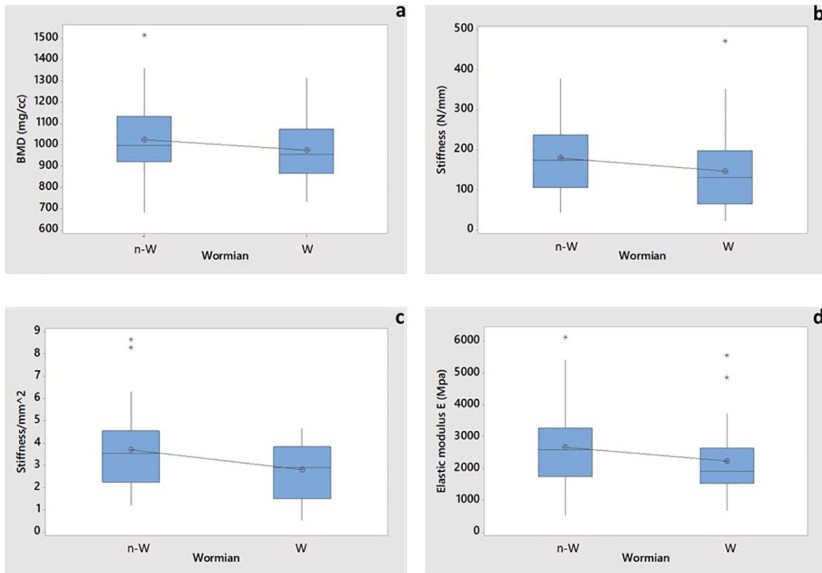

**Fig 8. Boxplot of 2-sample t-test for assessing the influence of Wormian bone on.** a) BMD (p-value 0.252), b) Stiffness S (p-value 0.221), c) Stiffness normalized with respect to the cross-sectional area (p-value 0.038) and d) Young's moduli E (p-value 0.198). Asterisks represent the outliers. N-W indicates no-Wormian samples, W indicates Wormian samples.

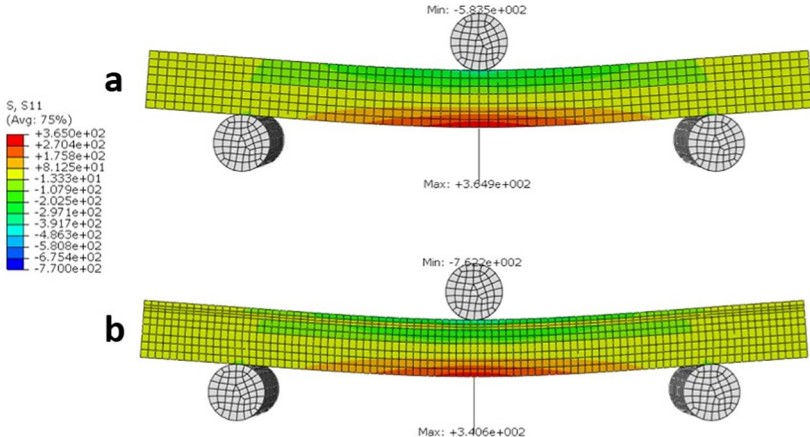

**Fig 9. Longitudinal stress S11 (MPa) distribution for a) non-Wormian bone model, b) Wormian bone model.** The maximum longitudinal stress for the Wormian model is ~7% lower than the non-Wormian model.

non-Wormian group was 19% larger (2.67 GPa) compared to the Wormian group (2.23 GPa) (Table 2, Fig 8d). This result, however, was not statistically significant (p-value = 0.198).

**Numerical results.** The stress distribution S11 (Sx) for both the non-Wormian and Wormian sample is illustrated in Fig 9. This shows that the maximum longitudinal stress S11 of the Wormian model is ~7% lower with respect to the non-Wormian model. Qualitatively, it can also be observed that the area of maximum stress is slightly larger for the non-Wormian samples. This indicates a wider distribution of the longitudinal stresses in the Wormian specimens, which would make those samples more resistant to bending loads.

Similarly, the strain distribution (LE11 or LEx) in Fig 10, indicates that the maximum strain is 9% lower in the Wormian bone model when compared to the non-Wormian model. This would cause the Non-Wormian model to fail earlier under the same loading conditions. As with the longitudinal stress, qualitatively, the strain is also more widely distributed within the Wormian model when compared to the non-Wormian model. This disperses the failure zone across a wider area instead of concentrating the deformation in a small area.

Values of bending Stiffness were computed for each model (non-Wormian and Wormian bone) by calculating the slope of the load versus displacement data shown in Fig 11. The

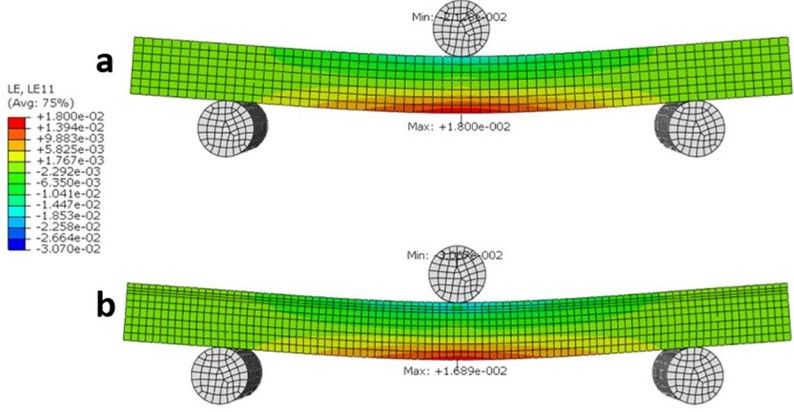

**Fig 10. Longitudinal strain LE11 distribution for a) non-Wormian bone model, b) Wormian bone model.** The maximum longitudinal strain is for the Wormian model is ~9% lower than the non-Wormian model.

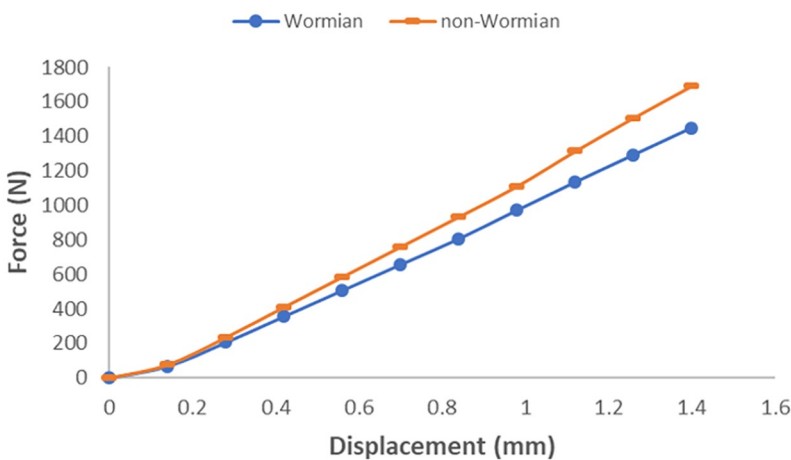

**Fig 11. Load-displacement curve for the loading probe for parametric non-Wormian (dash) Wormian (circle) models.**

bending Stiffness of the Non-Wormian model, calculated as 1299.12 N/mm, is 14.6% higher than the one of the Wormian model, estimated as 1109.90 N/mm. This shows that the higher compliance of the Wormian bone sample is attributable to the different material properties within the structure, as a result of the presence of Wormian bone. Encouragingly, the variation in the stiffness calculated from simulations (14.6%) is close to that observed for the experimental tests (19%).

## Discussion

Upon examination of equine parietal bone, it was found that Wormian bone was present in 70% of the examined population. The subjects employed for this work were free of any bone-related pathologies and hence, this is evidence that Wormian bone is a common trait present in the healthy horse skull. This appears to also be in case in humans [36] where up to 80% incidence has been reported in a Chinese population [2].

In the current study, no relationship could be established between the presence of Wormian bone and the sex, nor the age of the subject. As Wormian bone is equally likely to be present at any age, it implies that its existence is not associated with bone growth or bone development, contrary to what has been claimed in [1]. However, due to a low number of subjects (N = 10), further studies are required to verify this hypothesis.

Experimental results revealed that the presence of Wormian bone was found not to significantly impact BMD (Table 2). It would be expected that Wormian bone's apparent resemblance in material properties to those of trabecular bone, which is known to have lower density than cortical bone [23, 37], would cause a local reduction in BMD. However, it is likely that any local differences in BMD were not detected since the BMD was averaged over the entire sample. Wormian bone samples exhibited a lower (normalised) bending stiffness (P = 0.038) when compared to the samples without Wormian bone. While not statistically significant, our results showed a 14% lower Young's modulus for Wormian bone samples (P = 0.198). While use of the Euler-Bernouilli model to infer the Young's modulus implies a number of assumptions which are difficult to apply to biological samples (heterogeneous, irregular samples), most previous studies concerned with the flexural performance of cranial bone have also utilized the Euler-Bernoulli theory [27, 38–44], even when the curvature of the samples were very pronounced [45, 46]. However, in an attempt to account for the experimental uncertainties

present and to confirm our hypothesis that Wormian bone induces greater compliance in cranial bone samples, an idealised FE model of both Wormian and non-Wormian bone samples was created.

Encouragingly, the results of the FE simulations were consistent with the experimental results. Eliminating the geometric limitations associated with testing irregular cranial bone samples in 3-point bending supported our hypothesis that the inclusion of Wormian bone caused a decrease in the stiffness, meaning that Wormian bone is more compliant and less likely to fracture on impact. Additionally, a wider distribution of the longitudinal stresses and strains in the Wormian specimens was observed, which would make those samples more resistant to bending loads.

The present results confirm that the addition of an extra undulating layer of lower density material (Wormian bone) increases the damping behaviour of a sandwich structure, in this case, the cranial bone. The engineering design of sandwich panels aims for the reduction of vibration, noise and impact loads, dampening the deformations as well as the stresses and the strain rates [33–35]. The same logic can be applied to the presence of diploe throughout the neurocranium, which may exist to dissipate traumatic loads efficiently across the cranium, thereby reducing the risk of damage to the underlying and vulnerable neural tissue. This effect has been found elsewhere in nature, where an uneven and widespread presence of trabecular bone in the woodpecker skull contributes to dampening the effects of impact to the brain [47]. We believe that the random distribution of Wormian bone has a similar effect and may act as an additional design feature to dissipate traumatic stresses and strains and protect the underlying neural tissue. While the evolution of Wormian bones is not addressed here, these results support the theory that their development is more likely related to natural selection than to functional adaption of the bone [21].

## Conclusions

According to the results of the current study, the occurrence of Wormian bone in equine samples is 70% and is neither linked to the age nor sex of the subject. However, due to a low number of subjects (N = 10), further studies are required to verify this conclusively.

The mechanical behaviour of Wormian-containing samples of equine parietal bone and comparison control samples that lacked Wormian bone were examined using three-point bending tests and idealised three-dimensional finite element simulations. The experiments showed that the existence of Wormian (intrasutural) bone reduced the normalised bending stiffness (P = 0.038) when compared to samples without Wormian bone. Furthermore, while not statistically significant, it appears that the presence of Wormian bone also reduced the bending stiffness (P = 0.221) and elastic modulus (P = 0.198) of the bone, making it less stiff and hence more flexible. An idealised FE model confirmed this observation by demonstrating a more extended distribution of stress across the cross-section of the samples and a reduction of ~7% in the value of the maximum stress. This was also observed for the longitudinal strain (~6%). Furthermore, the idealised FE model confirmed that the stiffness of the non-Wormian model is 14.6% lower than the Wormian model.

For the first time, the current study proposes that the presence of Wormian bone could contribute to the protection of the brain in an impact. The reduction in stiffness, together with the wider distribution of the stresses and strains act in a similar manner to cancellous bone, by dampening the effect of impacts to the brain [47] and reducing the likelihood of fracture. We consequently conclude that the presence of these Wormian (intrasutural) bones act as a protective mechanism to shield the brain from damage when impacts to the skull occur. The analysis of the biomechanical contribution of Wormian bone presented here offers new insights

into their evolution and may provide reasonable explanation for the variation in incidence across species and human populations.

## Supporting information

**S1 Fig. Cross-section slice of an equine cranial bone sample showing a profile line reporting the grey values for pixels in the trabecular bone zone.**
(TIF)

**S2 Fig. Cross-section slice of an equine cranial bone sample showing a profile line reporting the grey values for pixels in the Wormian (Intrasutural) bone area.**
(TIF)

**S3 Fig.**
(PDF)

## Acknowledgments

The authors would like to acknowledge Peter O'Reilly of Trinity College Dublin and both Pat Kearns and Louise Brennan, of the UCD School of Veterinary Medicine.

## Author Contributions

**Conceptualization:** Lilibeth A. Zambrano M., Aisling Ní Annaidh.

**Formal analysis:** Lilibeth A. Zambrano M., David Kilroy, Arun Kumar, Michael D. Gilchrist.

**Funding acquisition:** Aisling Ní Annaidh.

**Investigation:** Lilibeth A. Zambrano M.

**Methodology:** Lilibeth A. Zambrano M., Michael D. Gilchrist, Aisling Ní Annaidh.

**Supervision:** Michael D. Gilchrist, Aisling Ní Annaidh.

**Validation:** David Kilroy, Arun Kumar, Aisling Ní Annaidh.

**Writing – original draft:** Lilibeth A. Zambrano M.

**Writing – review & editing:** David Kilroy, Arun Kumar, Michael D. Gilchrist, Aisling Ní Annaidh.

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
