## [Decision Letter · Decision Letter 0]

12 Jan 2021

PONE-D-20-32080

The presence of Wormian bones increases the fracture resistance of equine cranial bone

PLOS ONE

Dear Dr. Ni Annaidh,

Thank you for submitting your manuscript to PLOS ONE. After careful consideration, we feel that it has merit but does not fully meet PLOS ONE’s publication criteria as it currently stands. Therefore, we invite you to submit a revised version of the manuscript that addresses the points raised during the review process.

The reviewers have suggested adding a brief section or sentences describing the broader relevance of your work to the study of the biomechanics of the neurocranium.

We look forward to receiving your revised manuscript.

Kind regards,

John Leicester Williams, Ph.D.

Academic Editor

PLOS ONE

Journal Requirements:

2. Please add the following text from your manuscript to the Ethics Statement: 'Bone samples from equine parietal bones were collected from ten fresh-frozen heads (3 male and 7 female, 4-29 years old). These horses were obtained from an abattoir and were free  from  any  metabolic  or  bone-related  disease.  Due  to  the  origin  of  the  samples,ethical approval was not required for this work.'

Reviewers' comments:

Reviewer's Responses to Questions

**Comments to the Author**

1. Is the manuscript technically sound, and do the data support the conclusions?

Reviewer #1: Yes

Reviewer #2: Yes

2. Has the statistical analysis been performed appropriately and rigorously? 

Reviewer #1: Yes

Reviewer #2: Yes

3. Have the authors made all data underlying the findings in their manuscript fully available?

Reviewer #1: Yes

Reviewer #2: Yes

4. Is the manuscript presented in an intelligible fashion and written in standard English?

Reviewer #1: Yes

Reviewer #2: Yes

5. Review Comments to the Author

Reviewer #1: This manuscript is an interesting examination of the presence and function of Wormian or inter-sutural bones in the horse skull. While considerable attention has been devoted to such bones in humans, little work has been directed at the extent to which Wormian bones are observed in other mammalian taxa. What's particularly nice about this manuscript is that the authors use microCT to characterize morphological variation in Wormian bones and then complement such data with tests of the mechanical properties of the bony elements. Thus, my comments and criticism are largely restricted to the need to tie in the current findings of this study into a larger body of research on the biomechanics and function of the neurocranium.

Major Comments:

Unlike other regions of the skeleton such as the limbs and jaws, the neurocranium in mammals is characterized by low levels of in vivo bone strain as well as the absence of load-induced bone formation or dietary plasticity (e.g., Hylander et al., 1991, Am. J. Phys. Anthropol.; Hylander & Johnson, 1997, Am. J. Phys. Anthropol.; Ravosa et al., 2010, Anat. Record; Franks et al., 2016, Anat. Record; Franks et al., 2017, Zoology). Whereas safety factors to failure are uniform for mammalian jaws vis-a-vis masticatory stresses (much as the case for limb elements and locomotor stresses), the circumorbital region and calvarium are considerably overbuilt to resist routine biting/chewing stresses (Ravosa et al., 2010). Hylander and colleagues have argued that neurocranial bones are designed to counteract infrequent, traumatic loads that could otherwise inflict damage to the underlying brain. Accordingly, bone formation in the neurocranium may be buffered to variation environmental/mechanical stimuli so as to ensure that sufficient bone exists to resist traumatic loads (rather than masticatory forces) (Ravosa & Kane, 2017, Zoology). Thus, the authors' current findings regarding the greater compliance to external stresses afforded by Wormian bones is wholly consistent with the evidence from prior in vivo bone strain and adaptive plasticity work in mammalian crania. This, as well as the similar presence of diploe throughout neurocranial bones in humans, is yet another potential design feature that may exist to dissipate traumatic cranial loads that might otherwise injure subjacent neural tissues. Arguably, this paper would be strengthened significantly if the authors were to link their findings to this larger body of biological research on craniofacial design and biomechanics.

Minor Comments:

The authors observations about the presence of sandwich bone or diploe should be related to the similar presence in humans of hard-tissue organization where cortical bone along the periosteal and endosteal surfaces has intervening diploe or spongy bone in the neurocranium.

There appear to be formatting inconsistencies in the references where article titles sometimes capitalize all of the nouns and sometimes not. There's also some variability in journal citation formatting.

Reviewer #2: In general, this article is a well-executed study of the biomechanical effects of Wormian bones on biomechanical properties of cortical vault bone. Usually Wormian bones are analyzed as a non-metric skeletal trait with a genetic background. Because their prevalence varies across populations, analysis in humans specifically focuses on their presence, absence, or number as a phylogenetic signature. This paper is interesting in that it instead focuses on their contribution to cranial mechanical integrity. The finding that Wormian bones can reduce the risk of mechanical failure is novel and significant. This has implications for studying the evolutionary origin and prevalence of Wormian bones in different species.

Lines 45-74: This introduction, while thorough, is overly long and extraneous to the matter at hand. I would suggest condensing it to focus specifically only on the process that produces Wormian bones.

Lines 75-96: This is an appropriate level of introduction to the topic if combined with a condensed version of the preceding paragraphs. The justification for the study is clear and the choice of animal model is justified.

As the origins of Wormian bones is important, it may be useful to cite the literature demonstrating that its prevalence exhibits genetic variation. Ie: the prevalence may be higher in some populations. This is eluded to in text, but can be stated directly.

Line 99: Add is after hypothesis.

Lines 138-139 : change was to were. Wormian bones is plural.

Line 320: remove the before BMD

Line 336 “By eliminating the geometric limitations associated with testing irregular cranial bone samples in 3 point bending, it was proven” would read better as “Eliminating the geometric limitations associated with testing irregular cranial bone samples supported our hypothesis that…”

Line 359: stiffness should not be capitalized.

In the conclusion or introduction, I would appreciate a few sentences on the potential significance of the authors hypothesis. What contribution will these findings make to the field of skeletal biology? It seems to me that analyzing the biomechanical contribution of Wormian bones offers new insight into their evolution and prevalence across vertebrate species/ populations. A brief discussion of how this research advances the field would improve the article.

In general, the figures are well presented and easy to understand. It may be helpful to present individual data points in box-plots, but this is not necessary.

6. PLOS authors have the option to publish the peer review history of their article (what does this mean?). If published, this will include your full peer review and any attached files.

Reviewer #1: No

Reviewer #2: **Yes: **Benjamin Osipov

---

## [Author Response · Author response to Decision Letter 0]

16 Feb 2021

Dear Editor,

all reviewer comments have been addressed in the 'response to reviewers' document.

Kind regards,

Assoc. Prof. Aisling Ni Annaidh

---

## [Decision Letter · Decision Letter 1]

19 Mar 2021

The presence of Wormian bones increases the fracture resistance of equine cranial bone

PONE-D-20-32080R1

Dear Dr. Ni Annaidh,

We’re pleased to inform you that your manuscript has been judged scientifically suitable for publication and will be formally accepted for publication once it meets all outstanding technical requirements.

Kind regards,

John Leicester Williams, Ph.D.

Academic Editor

PLOS ONE

Additional Editor Comments (optional):

Reviewers' comments:

Reviewer's Responses to Questions

**Comments to the Author**

1. If the authors have adequately addressed your comments raised in a previous round of review and you feel that this manuscript is now acceptable for publication, you may indicate that here to bypass the “Comments to the Author” section, enter your conflict of interest statement in the “Confidential to Editor” section, and submit your "Accept" recommendation.

Reviewer #1: All comments have been addressed

2. Is the manuscript technically sound, and do the data support the conclusions?

Reviewer #1: Yes

3. Has the statistical analysis been performed appropriately and rigorously? 

Reviewer #1: Yes

4. Have the authors made all data underlying the findings in their manuscript fully available?

Reviewer #1: Yes

5. Is the manuscript presented in an intelligible fashion and written in standard English?

Reviewer #1: Yes

6. Review Comments to the Author

Reviewer #1: The authors have done a nice job of broadening the implications of their study to related experimental work on the functional significance of cranial vault architecture.

7. PLOS authors have the option to publish the peer review history of their article (what does this mean?). If published, this will include your full peer review and any attached files.

Reviewer #1: No

---

## [Editor Report · Acceptance letter]

8 Apr 2021

PONE-D-20-32080R1 

The presence of Wormian bones increases the fracture resistance of equine cranial bone 

Dear Dr. Ni Annaidh:

I'm pleased to inform you that your manuscript has been deemed suitable for publication in PLOS ONE. Congratulations! Your manuscript is now with our production department. 

Kind regards, 

on behalf of

Dr. John Leicester Williams 

Academic Editor

PLOS ONE